# Design and Implementation of Machine Vision-Based Quality Inspection System in Mask Manufacturing Process

**Minwoo Park** and **Jongpil Jeong** *

Department of Smart Factory Convergence, Sungkyunkwan University, 2066 Seobu-ro, Jangan-gu, Suwon 16419, Gyeonggi-do, Korea; minwoo.alvin.park@gmail.com
* Correspondence: jpjeong@skku.edu; Tel.: +82-31-299-4267

**Abstract:** With the advent of the 4th Industrial Revolution, research on anomaly detection in the manufacturing process using deep learning and machine vision is being actively conducted. There have been various attempts to innovate the manufacturing site by adopting advance information technologies such as machine vision, machine learning, and deep learning in many manufacturing processes. However, there have been no cases of designing and implementing these technologies at the mask manufacturing site, which is essential to tackle COVID-19 pandemic. The originality of this paper is to implement sustainability in the mask manufacturing environment and industrial eco-system by introducing the latest computer technology into the manufacturing process essential for pandemic-related disasters. In this study, the intention is to establish a machine vision-based quality inspection system in actual manufacturing process to improve sustainable productivity in the mask manufacturing process and try a new technical application that can contribute to the overall manufacturing process industry in Korea in the future. Therefore, the purpose of this paper is to specifically present hardware and software system construction and implementation procedures for inspection process automation, control automation, POP (Point Of Production) manufacturing monitoring system construction, smart factory implementation, and solutions. This paper is an application study applied to an actual mask manufacturing plant, and is a qualitative analysis study focused on improving mask productivity. "Company A" is a mask manufacturing company that produces tons of masks everyday located in Korea. This company planned to automate the identification of good and defective products in the mask manufacturing process by utilizing machine vision technology. To this end, a deep learning and machine vision-based anomaly detection manufacturing environment is implemented using the LAON PEOPLE NAVI AI Toolkit. As a result, the productivity of "Company A"'s mask defect detection process can be dramatically improved, and this technology is expected to be applied to similar mask manufacturing processes in the future to make similar manufacturing sites more sustainable.

**Keywords:** machine vision; quality inspection; quality assurance; anomaly detection; machine learning; deep learning; smart factory

## 1. Introduction

Entering the Post-COVID (Coronavirus Disease) era, creating a sustainable production environment across all industries has emerged as a top priority. COVID-19 pandemic has created surge demand for essential healthcare equipment along with the requirement for advance information technologies applications [1]. Health masks have emerged as a critical necessity as fine dust continues to gradually increase since the 2000s. According to AirVisual's 2018 World Air Quality Report, South Korea has the second highest concentration of ultrafine dust among OECD (Organization for Economic Cooperation and Development) member countries. In particular, 76.5% of South Koreans check the fine dust information daily and the need for masks is increasing due to issues such as yellow dust, MERS (Middle East Respiratory Syndrome), and COVID-19.

Quality assurance of products are considered one of the most important inspection factors in the manufacturing process, including the textile industry. Textile product quality is seriously degraded by defects. Textile fabrics exhibit uniform patterns and texture properties in both horizontal and vertical directions. The damaged regions, which disfigure the patterns and the texture, are called fabric defects [2]. Failure to defect detection early in the process costs time, money, and consumer satisfaction. Thus, early and accurate fabric defect detection is an important phase of quality control [3]. Textile defect detection is a critical step in quality control in textile manufacturing because if there are defects in the textile fabric, its market price is reduced by more than half. That's why Automatic fabric defect detection is considered to be of great interest for detection of different kinds of defects like hole, slub, oil stains, etc. [4]. The old way of carrying out defect detection is to assign a human inspector to look for flaws to ensure quality. Manual inspection is time consuming, and the level of accuracy is unsatisfactory to meet the present demand of the highly competitive international market [3]. Therefore, the expected quality cannot be maintained with manual inspection. A computer vision-based fabric defect inspection system is the solution to the problems caused by manual inspection because machine vision significantly improves the efficiency, quality, and reliability of defect detection [5]. Automated fabric defect inspection system has been attracting extensive attention of researchers across the globe for years [6].

In the case of the existing mask manufacturing process of "Company A", automation was carried out only during the fabric input and production process. On the other hand, in the QA (Quality Assurance) process of finished products, it went through a manual process to distinguish between good and defective products through human eyes and hands. As a result, time and cost were extremely inefficient. To address this problem, we aim to dramatically increase the productivity of "Company A"'s mask defect detection process by designing and implementing a machine vision-based inspection system in the mask manufacturing process. To improve the performance of distinguishing good and defective products in the field through AI machine vision solutions, LAON PEOPLE NAVI AI Toolkit is used. Since its inception in 2010, LAON PEOPLE has been providing machine vision solutions for the manufacturing and logistics automation fields of leading companies in Korea and abroad. This program enabled the innovation of the manufacturing process with Korea's first artificial intelligence machine vision inspection software in 2016 and helped to use a variety of machine vision products according to the environment including; AI inspection solution packages, 2D/3D cameras, smart cameras, and thermal imaging cameras. This study is conducted as an application study for a real-world factory for performance improvement with qualitative analysis. The composition of this paper is as follows; Section 2 deals with Computer Vision, Machine Vision-Based Quality Inspection, Machine Learning for manufacturing process, and Remaining Useful Life Prediction, Section 3 explains Sustainable Smart Factory Mask Manufacturing Process, Section 4 deals with Implementation and results, Section 5 explains the Conclusion.

## 2. Related Works

### 2.1. Computer Vision

Computer Vision is an area that studies how to extract meaningful information from still images or moving images using a computer. In the case of the existing neural network, all neurons in adjacent layers are combined. CNN (Convolutional Neural Network) adds a pooling layer to this and uses the Affine-Softmax combination as it is in the last output layer by connecting to the flow of Conv-ReLU-(Pooling) [7]. Padding refers to a method of padding around data with specific values before performing a convolution operation. Without padding, it becomes smaller every time the convolution operation is performed, and the operation cannot be performed at any moment. CNNs have gained prominence in the research literature on image classification over decade. One shortcoming of CNNs is their lack of generalizability and tendency to overfit when presented with small training sets. Augmentation directly confronts this problem by generating new data points providing

additional information. That's why Image Data Augmentation is required in order to implement CNN [8].

When training is performed without augmentation, only the image of the train data used for training is detected well. That is, there may be an overfitting to detect fabric defects on gray and checked cloths with a detection error rate of less than 5% [9]. Data augmentation is a technique for generating multiple data with one data. Its purpose is to augment limited data to increase performance and solve overfitting problems. Image Data Augmentation techniques are divided into Basic Image Manipulation, Deep Learning Approach, and Meta Learning. Basic Image Manipulation is subdivided into Geometric Transformation, Color Space Transformation, Kernel Filter, Random Erasing, Mixing Image, etc. Deep Learning Approach is subdivided into Feature Space Augmentation, Adversarial Training, GAN Data Augmentation, and Neural Style Transfer. Finally, in the case of Meta Learning, it is subdivided into Neural Augmentation, AutoAugment, and Smart Augmentation [10].

### 2.2. Machine Vision-Based Quality Inspection

When an operator manually performs a quality inspection in the manufacturing process, their eyes become fatigued over time. For this reason, if the test is performed on a subjective basis by humans, sustainable quality control cannot be maintained. According to relevant statistics, the traditional fabric inspectors could inspect up to 200 sheet fabrics in 1 h [11]. This is very time-consuming in the production chain. In recent years, many scholars have explored corresponding application of computer vision technology to the detection of fabric defects [12]. Due to the advantages of high accuracy, low costs, and nondestructive testing, the object detection-based technologies are applied extensively in the various domain. With the development of such computer vision technology, many manufacturing companies use computer vision to identify defects in products in real time. No matter how small the product is, the computer can process images or videos of the finished product to identify dozens of defects. Product quality inspection is crucial in the manufacturing process. Extensive research has been conducted to detect defective products using image analysis and machine vision accounts for the largest portion [13]. When various sensors are used together, defective products can be detected more accurately, but there are disadvantages in that additional resources are consumed and the cost of using the sensor increases. Therefore, in recent years, the development and use of a machine vision-based defect detector using a camera has become more prevalent [5]. As the 4th Industrial Revolution accelerates, AI functions began to be added to the detection of defective products using image processing. Countless studies have resulted in the development of deep learning and machine vision-based defect detection software. With the use of the Keras open source library, defects based on images of normal products were determined and defects based on probability distribution were located [14].

In general, defect detection networks can be divided into two types: two-stage detection and one-stage detection [15]. One-stage detection algorithm is an end-to-end detection scheme. The feature is extracted directly from the convolution neural network. The major one-stage method include YOLO (You Only Look Once), SSD (Single Shot Detector), RetinaNet and so on. RetinaNet is a neural network that uses two sub-networks with RestNet (Deep Residual Learning for Image Recognition)-FPN (Feature Pyramid Network) as the backbone [16]. On the other hand, the two-stage detector has high accuracy on defect location and recognition. Cha et al. developed a structural damage detection method based on Faster R-CNN (Regions with Convolutional Neuron Networks features) to detect five types of surface damages [17]. Sifundvolesihle et al. studied the development of a real-time machine vision system for functional textile fabric defect detection using a deep YOLOv4 (Optimal Speed and Accuracy of Object Detection) model [2]. According to the methodology they tried, the YOLOv4 model was applied to the textile fabric in the order of Data Preprocessing, Data Augmentation, Data Labeling, and Defect Localization.

### 2.3. Anomaly Detection for Manufacturing Process

Anomaly detection, a.k.a. outlier detection or novelty detection, is referred to as the process of detecting data instances that significantly deviate from the majority of data instances [18]. Anomaly detection for industrial processes is essential in industrial process monitoring and is an important technology to ensure production safety [19]. Manufacturing companies use cameras and laser sensors to make the surface of the product or the condition of the product into data. By using this data, various studies have been conducted to automatically perform quality tests, such as statistical methodology, image processing methodology, and methodology using machine learning models. Statistical methodology helps manufacturers make meaningful decisions for workers or managers by collecting and analyzing data produced in the field to improve quality control and processes. Statistical methods help to correlate, organize, and interpret data, and statistical analysis shows the underlying patterns in a data set. Machine Learning is usually presented as an approach used for the inspection of smart manufacturing and impacts the quality control systems of industries [20]. Histogram analysis performs various statistical analysis including mean, geometric mean, standard deviation, and median. This analysis method is simple therefore it has been widely used for low-cost, low-level analysis in various problems [21]. Autocorrelation analysis measures the correlation between the image and the displacement vector from the image by using the pattern or texture of the product surface repeatedly, such as wood or textile products. The detection and timely evaluation of abnormalities in machine vision systems allow the industrial sector to make innovative leaps [22].

### 2.4. Remaining Useful Life Prediction

The concept of RUL (Remaining Useful Life) refers to the expected remaining lifespan of a component or system. Establishing a system that maintains and predicts the soundness of assets used in industrial sites is one of the tasks to be achieved for a smart manufacturing environment. Accurate RUL prediction is critical for prognostic and health management and for maintenance planning [23]. Most factories use large machines. Engineers attach vibration sensors to this mechanical equipment and measure vibration intensity to check the failure and remaining life of the equipment. By taking RUL into account, engineers can schedule maintenance, optimize operating efficiency, and avoid unplanned downtime. A recent study analyzed the vibration data used in the PHM (Prognostic and Health Management) IEEE 2012 Challenge, and found that the vibration of the machine in use becomes stronger as the lifespan expires [24]. There are CNN and LSTM (Long Short-Term Memory) models as neural network models for predicting the remaining lifespan. CNN is used to extract features, and LSTM is used to predict time series data. In this way, extracting features from a certain model is called deep feature extraction, and RMSE (Root Mean Square Error) is used as the RUL evaluation method. Kang et al. developed a novel RUL prediction approach that utilizes the principal component analysis (PCA) feature selection algorithm, grid search parameter optimization algorithm, and multi-layer perceptron (MLP) machine learning algorithm [25].

One of the approaches to reducing maintenance costs is known as preventive maintenance (PM) because fixing the production line after the breakdown can be more costly than conducting preventive maintenance ahead of the breakdown. Predictive maintenance (PdM) of production lines is important to early detect possible defects and thus identify and apply the required maintenance activities to avoid possible break-downs [25]. Breakdowns impact the performance and cost of production, and lead to a reduction of availability because of the costly maintenance period [26]. Recently, many researchers are focusing on research on predicting and preserving the lifespan of assets using machine learning. Machine Learning methods have been applied in different manufacturing areas and fault diagnosis is the most common application area of it. Luo and Wang applied random forest to identify the malfunction of robot arms by learning patterns from the torque sensor [27].

## 3. Sustainable Smart Factory Mask Manufacturing Process

### 3.1. The Problems with "Company A"'s Existing Mask Manufacturing Process and the Following Solutions

This section describes the process for analyzing the existing mask manufacturing process of "Company A" to find problems and suggest solutions. Figure 1 shows the existing workflow chart for each major process. In the Fabric Insertion stage, the fabric for mask manufacturing is supplied to the left, right, center, and nose wires. The Fabric Alignment Control stage is the process of balancing the left and right sides of the fabric and aligning the location of the fabric to ensure that the Nose Wire Insertion is properly implemented. In the Nose Wire Insertion and Wire Cutting step, the nose wire is properly cut and inserted into the fabric inserted in the previous step. Thereafter, 1st Ultrasonic Welding is performed, and the fabric is folded to perform 2nd Ultrasonic Welding. In the Fabric Cutting stage, when Ultrasonic Welding is finished, cutting is performed according to the shape of the mask, and unnecessary extra scrap is removed. Finally, good and defective products are classified through screening tests along with the removal of the defective products.

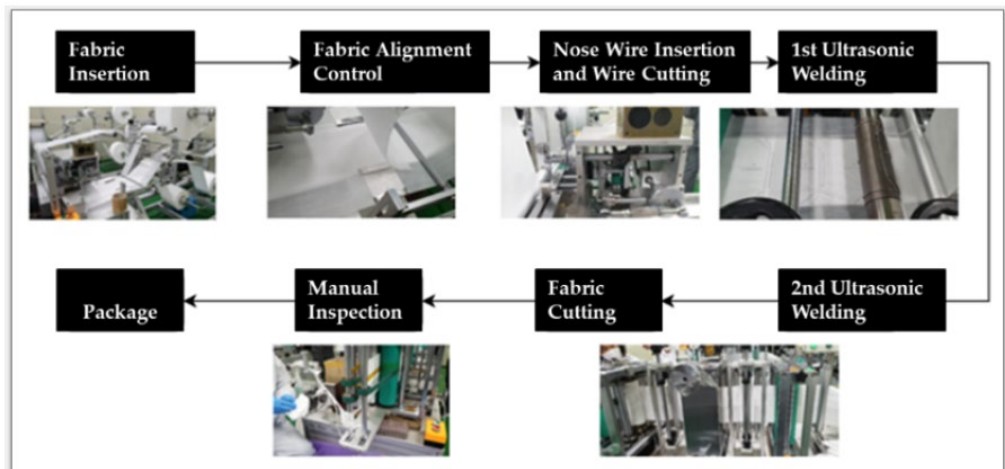

**Figure 1.** Workflow chart for each major process.

There is a problem in that spatial constraints for constituting sensors and machine accessories are large, and that they are designed using inverter motors rather than automation methods. In addition, there is no manufacturing management system such as production management and quality control because the production facilities are not computerized. Due to the lack of systematic production management, some masks are classified as good products even though they are in fact defective. Therefore, it is necessary to establish a production management system step by step in order to manage realistic product specifications and production.

There are following problems with the existing manufacturing process. The fabric tension control and alignment are not uniformed in the tension control process. Therefore, fabric derivation occurs because tension control and alignment are not unformed, and it means that the fabric is inserted into an inappropriate position due to the imbalance in the left and right positions of the fabric. In addition, a foreign object test on fabric is conducted by an inspector, but an internal inspection test was not possible. Foreign object test on fabric is a process of inspecting whether the fabric is contaminated by foreign substances such as dust and stains. Internal inspection test refers to an inspection process that checks whether there is contamination inside the fabric. Since the three fabrics of left, right, center overlap and are inserted, the factory operator can only check the fabric on the surface, so the lining part of the fabric cannot be visually inspected.

### 3.2. Machine Vision-Based Inspection Points on the Factory Floor to Solve Problems in the Maskmanu Facturing Process

In order to clearly identify these problems and create solutions for them, the following inspection points have been analyzed. The fabric tension control and alignment position are disproportionately formed due to irregular tension throughout the process, such as the Fabric Insertion part and the Ultrasonic Welding part, resulting in a defect. To solve this problem, machine vision-based inspection points are applied to the process. The decision is made to build a fabric inlet tension control system and make plans to build a supply power control system according to the weight of raw materials and a position control system through vision inspection or sensor feedback.

Figure 2 shows the inspection point of the nose wire fusion test. It is impossible to check whether the ultrasonic welding process is in progress in real time, and it is conducted by manually checking whether the product is defective. The quality of the inspection is not uniform due to the increase in fatigue caused by the inspector's long-term inspection. To solve this problem, the internal and external machine vision-based inspection points of the mask fabric are determined, and a foreign sub-stance inspection and fusion point inspection solution are introduced.

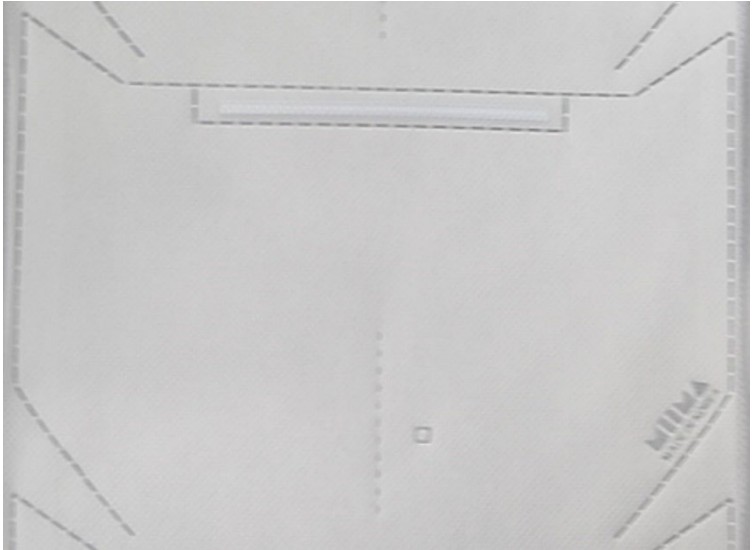

**Figure 2.** Nose wire fusion test.

Figure 3 shows the machine vision-based inspection point for foreign object test on fabric. Manufacturing automation has been established that can aggregate good and defective products according to inspection results.

### 3.3. Machine Vision-Based Quality Inspection System Architecture

Figure 4 shows the overall hardware equipment to implement machine vision-based quality inspection system. To automate the input fabric tension, a motor is attached to the input of the fabric so that the input tension can be adjusted. A machine vision test was conducted on the fusion state in ultrasonic welding. In the case of ear straps, left and right alignments were checked through sensors. The cutting drive type according to the chain rotation cycle is improved to an independent air cylinder type cutting process. Therefore, the nose wire insertion and cutting can be performed at a certain point in time to prevent the shaking of the chain connected to the nose wire cutter from affecting the entire facility.

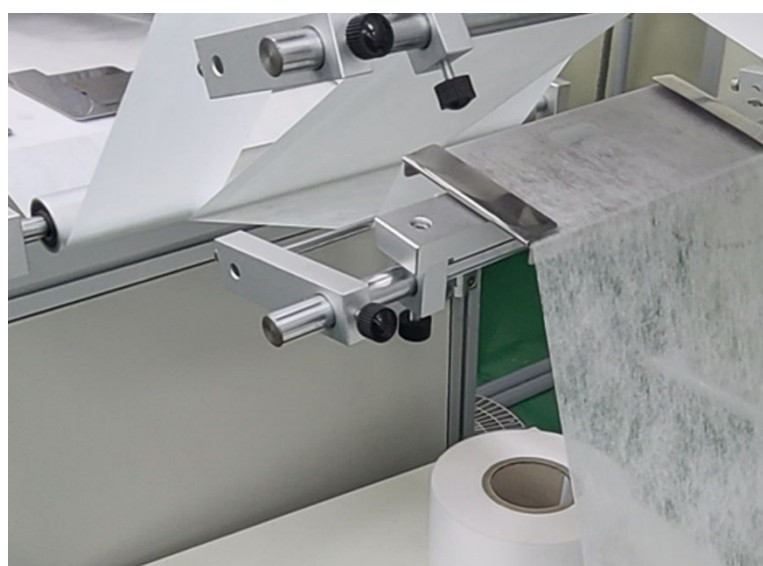

**Figure 3.** Foreign object test on fabric.

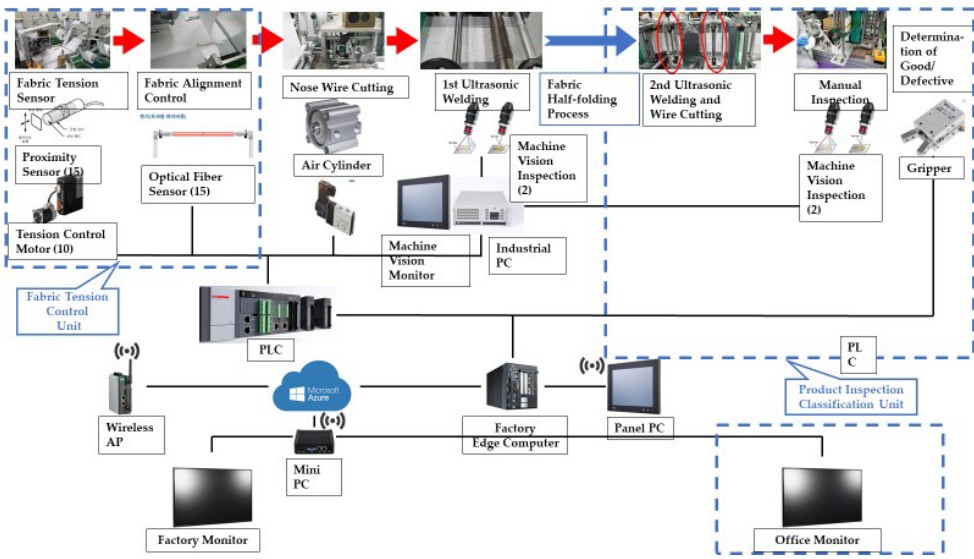

**Figure 4.** Hardware Architecture.

Figure 5 is Factory Layout and Equipment designed for machine vision-based quality inspection. To automate the inspection process that used to distinguish between good and defective products through visual inspection, machine vision is used to classify defective products through Mask Cutting and Ultrasonic Welding. In addition, it is made to automatically classify the results of the judgement of good and bad products through a automatic classification system. To collect images of good and defective products, three vision inspection points are set for Fabric Insertion, Ultrasonic Welding, and Fabric Cutting inspection. Images are collected so that vision images generated at the inspection points can be checked in the program regardless of both good and defective products.

For the sake of enabling facility control such as start, stop, and motor control of the fabric input part on the POP touch panel, it is implemented to check the Machine Vision shot image in the POP. Also, in the UI (User Interface) design, the facility images are expressed similarly to the actual facility. During the production status monitoring, the overall line of the process and the input status of each product are expressed as charts. In addition, the status of the facility and production status can be expressed on the dashboard. Through production defect management, the history of defects can be managed and analyzed. Quality management by period, quality management by work instruction,

and comprehensive quality management is possible. Figure 6 Shows the dashboard for monitoring effective mask products on the POP touch panel screen.

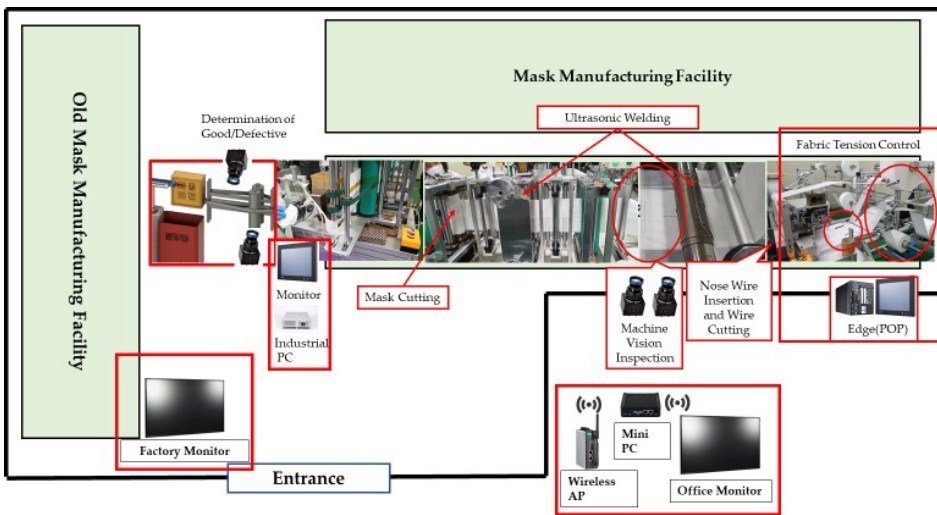

**Figure 5.** Factory Layout and Equipment.

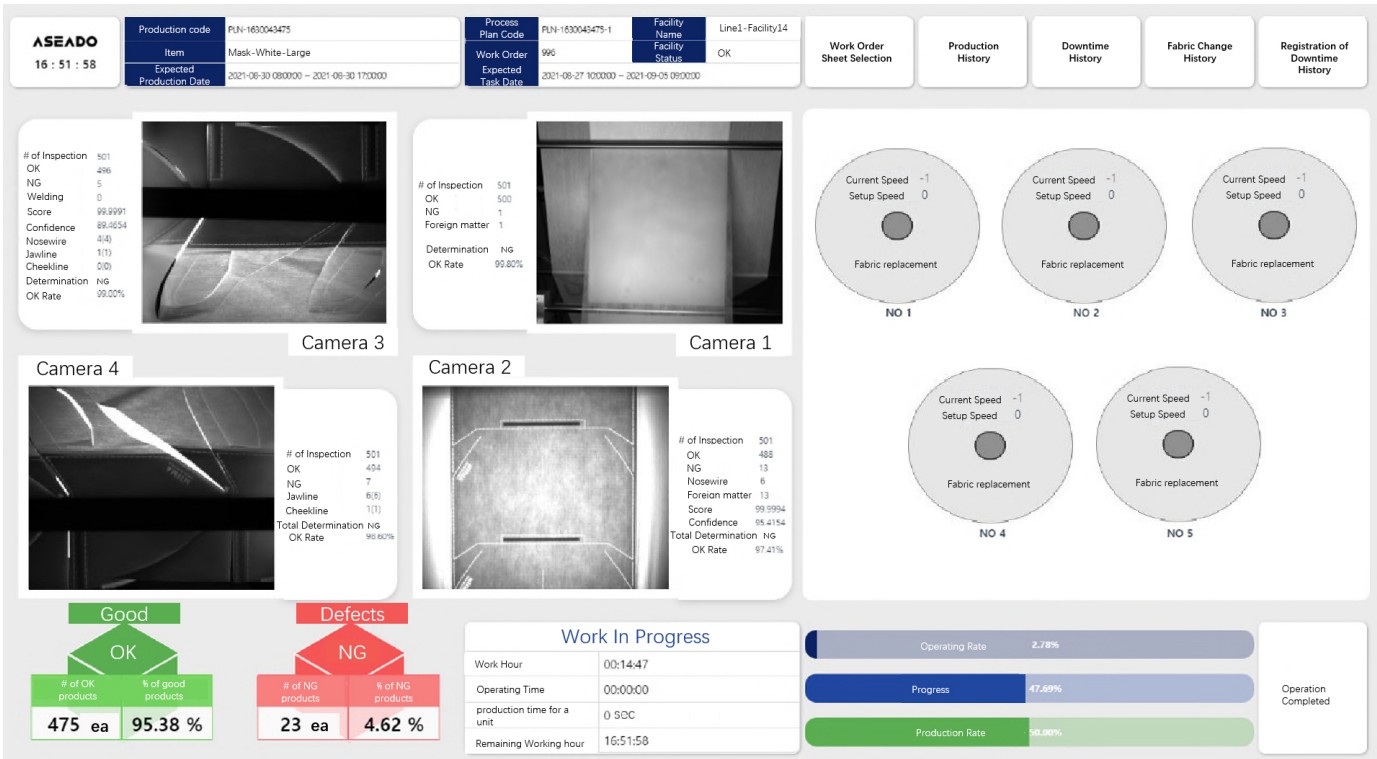

**Figure 6.** Dashboard for monitoring defective mask products.

In order to devise a method of displaying notifications when fabric shortage is expected, it has been confirmed that the smaller the remaining amount of fabric, the smaller the diameter of the fabric, the higher the speed of fabric release of the lift control device. When the speed of the fabric rises above a certain speed, an alarm rings to indicate the need for fabric replacement.

## 4. Implementation and Results

### 4.1. Hardware Specification and Network Configuration for Machine Vision-Based Inspection

Table 1 summarizes the Hardware Specification. The categories are classified into Hardware Equipment, Computing Equipment, Inspection Equipment, Dashboard, and Kiosk, and each quantity and specification were specified.

**Table 1.** Hardware Specification.

| Category | Item Name | Volume (ea) | Specification |
|---|---|---|---|
| Hardware Equipment | Edge Device | 2 | Intel Core 9th Generation 4G DDR4 Memory, 1TBHDD |
| Computing Equipment | Panel PC | 1 | Intel Haswell i5—4300U 1.9 GHz |
| Computing Equipment | Industrial PC | 1 | IPC610-I786 |
| Others | Big Size Monitor | 2 | 3840 × 2160@60 Hz |
| Others | Mini PC | 1 | Intel Core i5—8265/HD Graphics 620 |
| Others | High Brightness LED Lights | 5 | JL-F-D-230 × 200/180 |
| Others | Light Controller | 2 | JV504 |
| Others | POE Board | 1 | IntelGigabit POE 2ch |
| Others | Keyboard and Mouse | 1 | K400PLUS (Wire Keyboard) |
| Inspection Equipment | Camera | 4 | MV-CA050-10GM/Gige, 5M Pixel |
| Telecommunication Equipment | Wireless AP | 1 | AC4300 Wireless |
| Dashboard | Monitor | 1 | Alphascan AOC 24IPS77(24inch) |
| Kiosk | PLC | 1 | XGT/XGB [1] |

[1] The categories are classified into Hardware Equipment, Computing Equipment, Inspection Equipment, Dashboard, and Kiosk, and each quantity and specification were specified.

Figure 7 shows Network Configuration equipment. ipTIME, Edge Device, Wireless AP, Panel PC, Mini PC, Factory Monitor, and Office Monitor are required. The ipTIME PoE 8000 is an existing equipment at the mask manufacturing factory, and the part marked with a blue box represents the newly purchased equipment. The solid black line represents the Wired Network, and the dotted line represents the Wireless Network.

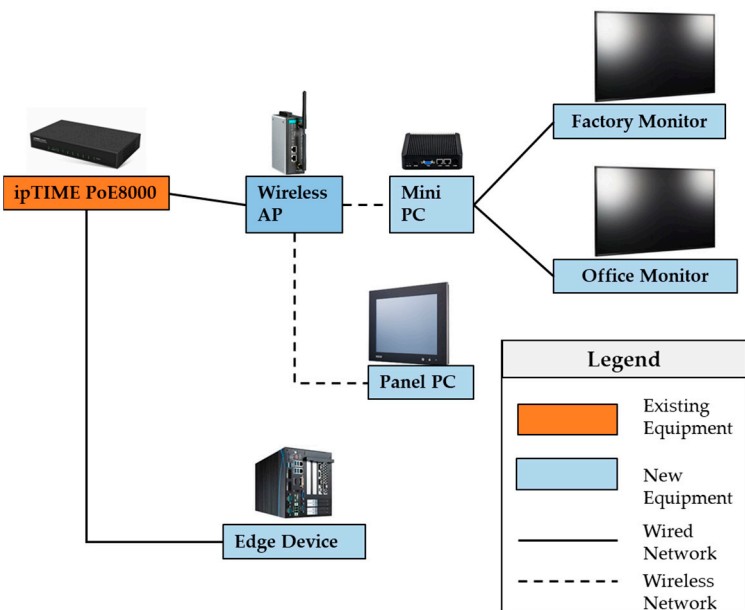

**Figure 7.** Network Configuration.

### 4.2. Communication Structure for Image Acquisition and Data Gathering Process

As shown in Figures 5 and 6, cameras were installed in each part of the machine vision defective product determination process. The cameras were connected to each port of the Power over Ethernet (PoE) LAN card with an Ethernet cable (cat.5 or higher). The Power

over Ethernet (PoE) refers to a system that reliably transmits data and power through an Ethernet cable. And the LAN card is a communication device in a computer that allows the computer to connect to a network and transmit data.

An Ethernet connection between the PLC and the Vision computer is performed to receive a photographing trigger through Socket communication. For image learning, machine vision learning is performed on the acquired image first. Then, a suitable model is created using NAVI Trainer. Next, learning is carried out by mixing unsupervised learning and supervised learning. The generated model is used to determine good and defective products for the acquired image. NAVI AI SDK USB is needed to build an image discrimination model. It must be connected to the vision computer by USB in order to distinguish between good and defective images.

Trigger.get is processed through an image acquisition and image determination process. Trigger.get was implemented with Socket communication. Trigger is an action that is automatically performed when the database meets a pre-determined condition or an action is performed. Triggers are useful for describing validity and integrity conditions for data in a database. HeartBeat is used to send machine vision results to MES. In a computer cluster, the HeartBeat network refers to a private network. Data is accumulated as a result from discriminating good and defective products through ma-chine vision. Vision.Result.Refresh received data from MES via MQTT in C++ code. Images are sent to MES via HTTP as Python code. Interval receives data from the MES in Python code through MQTT. Topic is company/mes/server and the Interval information is included in the refresh topic.

Table 2 summarizes the Data Gathering Process at a glance. The process goes through Fabric Tension Control, Cutting of the Nose Wire Insertion, and then Ultra-sonic Welding. Afterwards, it goes through the Cut the Fabric process after going through the Foreign Matter Test on Fusion and Lining, Fabric Half-folding Process, and 2nd Ultrasonic Welding. After the end of Process in the middle, Data Gathering continues in the order of Inspection of Fusion and Foreign Substance on the Outer Fabric and Entry Process.

**Table 2.** Data Gathering Process.

| Process | Equipment | Input | Gathering Data |
|---------|-----------|-------|----------------|
| Fabric Tension Control | Optical Sensor, Proximity Sensor | PLC Signal | Ensure the fabric is properly aligned with the inlet<br>Detect up/down and reverse rotation data errors of fabric |
| Cutting of the Nose Wire Insertion | Proximity Sensor | PLC Signal | A separate cutting cylinder operation control sensor from the existing sensor<br>Extraction of position data from a power transmission device using chain rotation |
| | | Ultrasonic Welding. | |
| Foreign Matter Test on Fusion and Lining | Machine Vision Camera | Ethernet RS485 | Ultrasonic Welding Inspection Data<br>Foreign material inspection data on the inside of the fabric |
| | (Process in the middle) Fabric half-folding process → 2nd Ultrasonic Welding → Cut the fabric. | | |
| Inspection of Fusion and Foreign Substance on the Outer Fabric | Machine Vision Camera | Ethernet RS485 | Foreign material inspection data<br>Mask appearance and welding inspection data<br>Distinguish between good and defective products among finished masks |
| | Edge Device | TCP/IP RS485 | PLC Control Data<br>Machine Vision Data<br>Production data |
| Entire Process | Panel PC | TCP/IP | Equipment Control Data<br>Defective Type, Daily Maintenance, Non-operation Type |
| | Mini PC | TCP/IP | Production data |
| | Machine Vision Camera | Automation | Vision defect inspection and automatic counting |

### 4.3. Database Management: Loading Image Data for Training, Classifying and Setting Types

Image data for the Front Fusion Line Inspection and the Back Fusion Line Inspection as well as the nose wire fusion test of Figure 2 and the foreign object test on fabric of

Figure 3 are obtained from the Machine Vision camera. Figure 8 shows the Front Fusion Line Inspection and Figure 9 shows the Back Fusion Line Inspection taken by the camera.

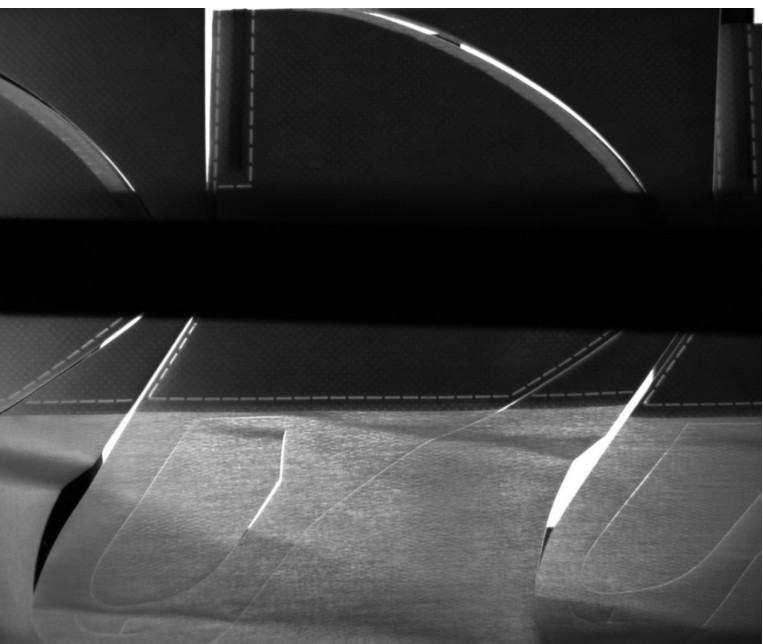

**Figure 8.** Front Fusion Line Inspection.

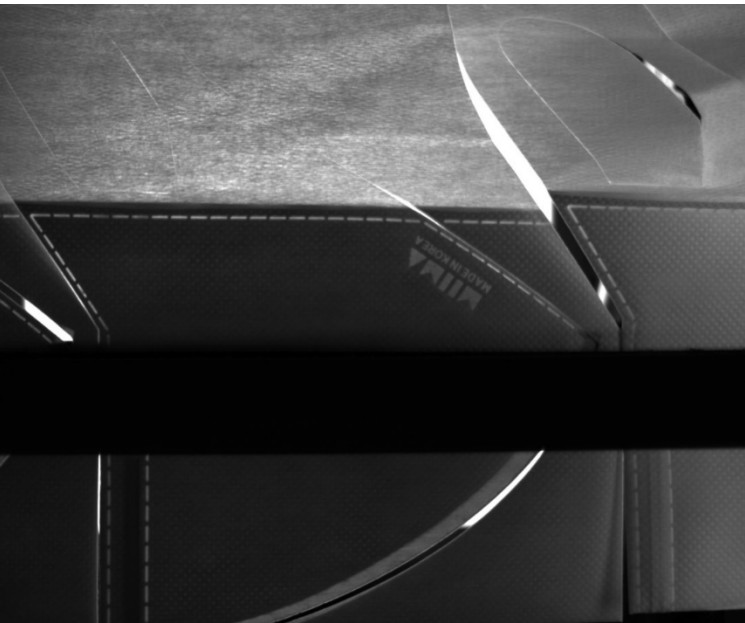

**Figure 9.** Back Fusion Line Inspection.

In the database stage, image data loading for training, type classification, and setting are managed. In the database creation step, a database that will be a training unit is added. For class configuration, classes are created for classification of types of images to be inspected. In the image classification step, addition/delete/movement is performed to classify the image of the class. Figure 10 is a screen that imports Good Image for AI Model training, and Figure 11 is a screen that imports Defect Image for training.

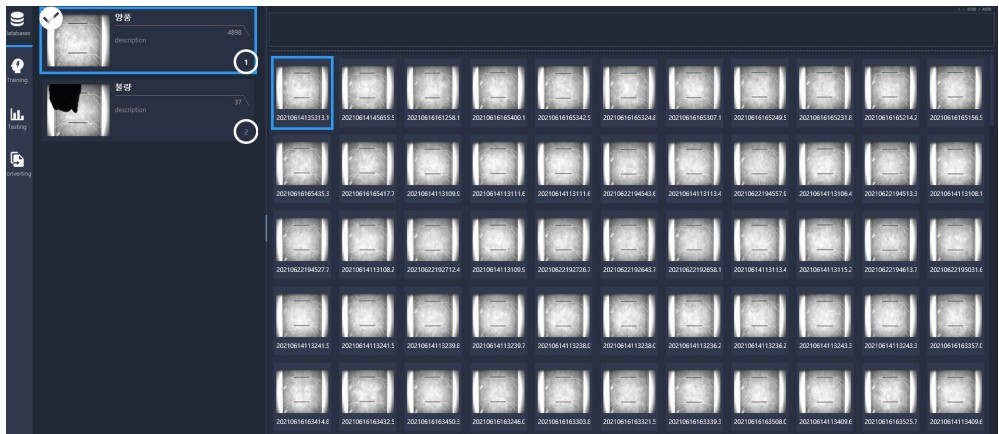

**Figure 10.** Good Image Import.

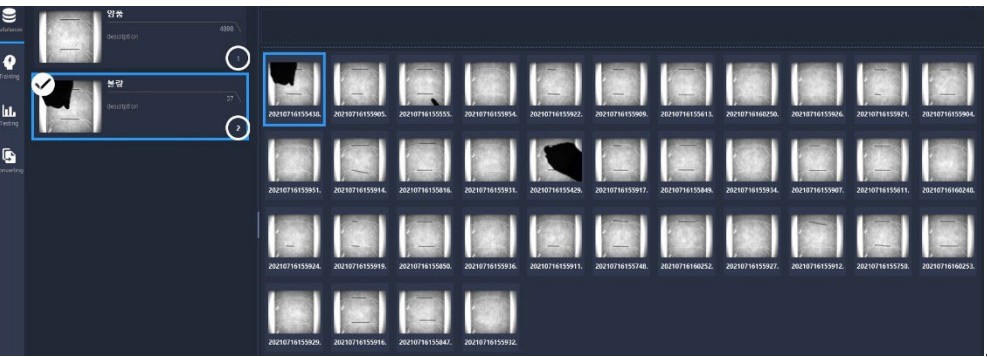

**Figure 11.** Defect Image Import.

Then, in the ROI designation step, a fixed region of interest of the retrieved images is designated and it also managed the defects of learning materials as a specific area and trained the selected area to identify the defect location. Figure 12 shows the ROI for Jaw Line, and Figure 13 shows the ROI for Cheek Line, respectively.

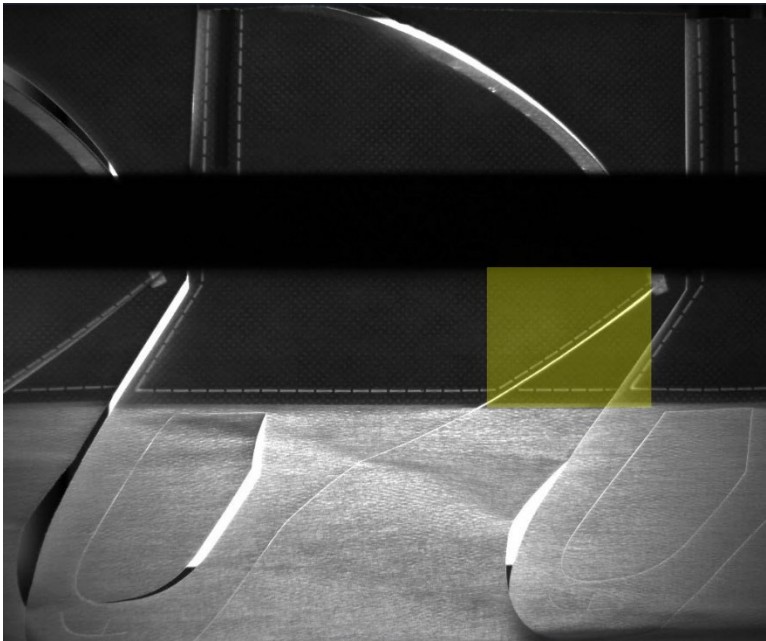

**Figure 12.** ROI for Jaw Line.

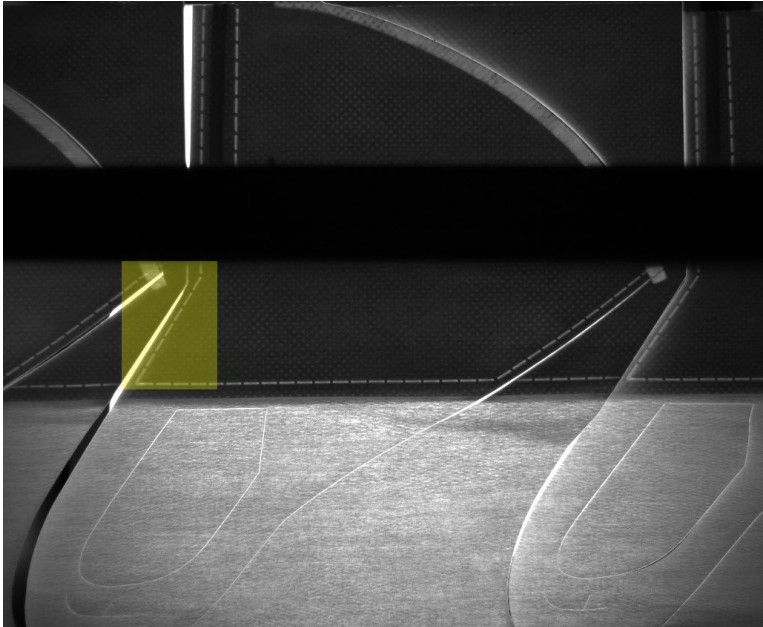

**Figure 13.** ROI for Cheek Line.

Figures 14 and 15 are screens that label the Detection Line, which will be the basis for determining the good and defective products of the mask, within the ROI area.

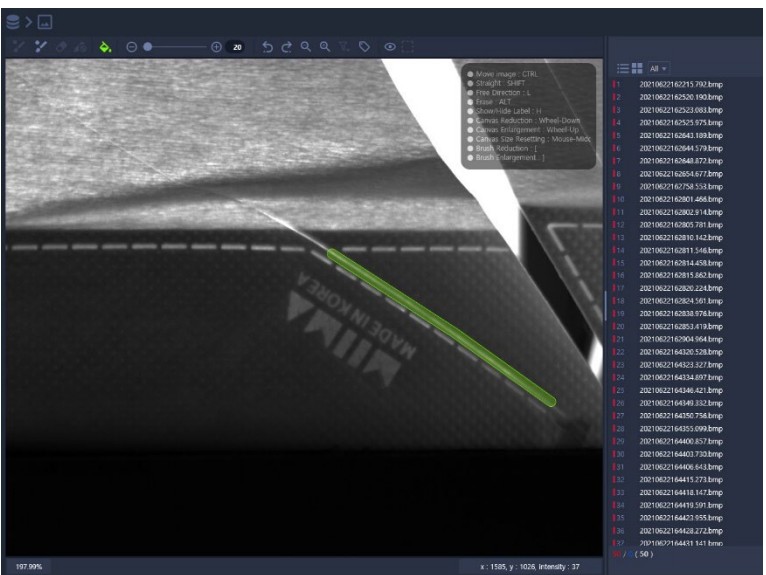

**Figure 14.** Detection Line for Jaw.

As shown in Figure 16, In the parameter designation step, parameters are adjusted according to the characteristics of the retrieved image data. Learning parameters were set through Solver Parameters. Max Iteration means the maximum number of learning iterations, Step Count means the size of increase per learning cycle, and Base Learning Rate means the learning speed. In general, the smaller the learning speed, the slower the learning speed, and the greater the accuracy. Gamma refers to the coefficient value involved in the Base Learning Rate. Random Count, Probability is associated with overfitting issues, and when learning, it is well detected for images used in learning and poor detection performance for images (test data) not used in learning. To prevent this, the learning image is augmented through Data Augmentation.

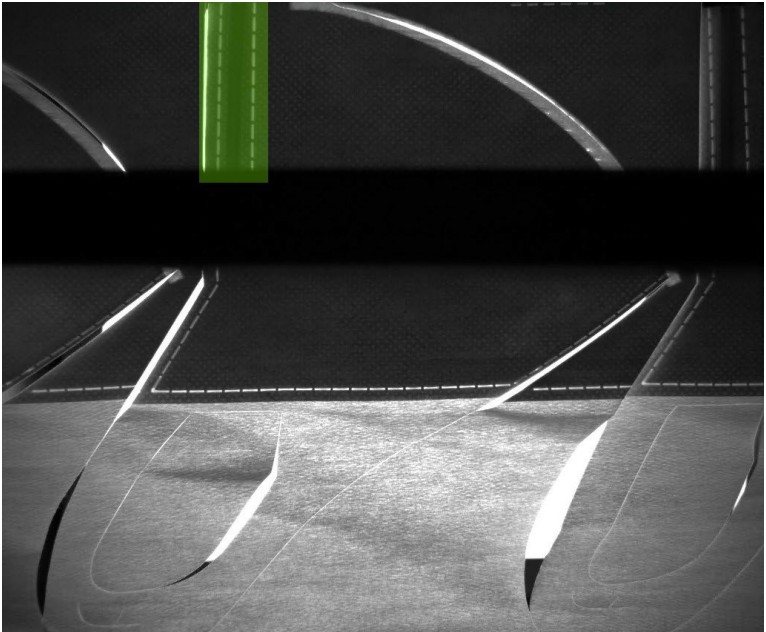

**Figure 15.** Detection Line for Nose Wire.

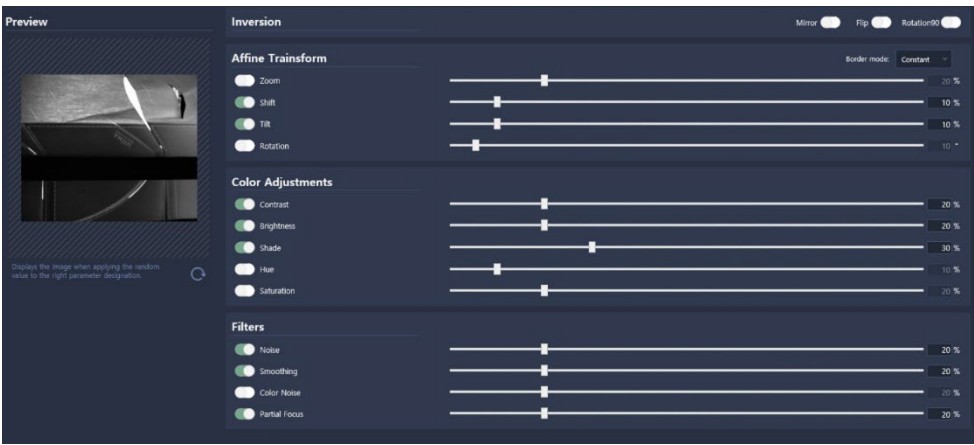

**Figure 16.** Parameter Setting and Training based on the Prepared Database.

In the Training test stage, the learning results are checked through the test and the corresponding results through a report. When learning begins, it checks the progress in real time. In the test report stage, after training, the test image is retrieved, and the test is performed at the same time to check the learning result. Figure 17 shows the AI Model Training Process. The learning graph refers to a graph representing the calculated value during real-time learning. Iteration refers to the repetition of current learning, Test Accuracy refers to the degree to which the prediction of the model currently being learned is accurate, Test Loss refers to the degree to which the prediction of the validation set is incorrect, and Train Loss currently learning model is incorrect.

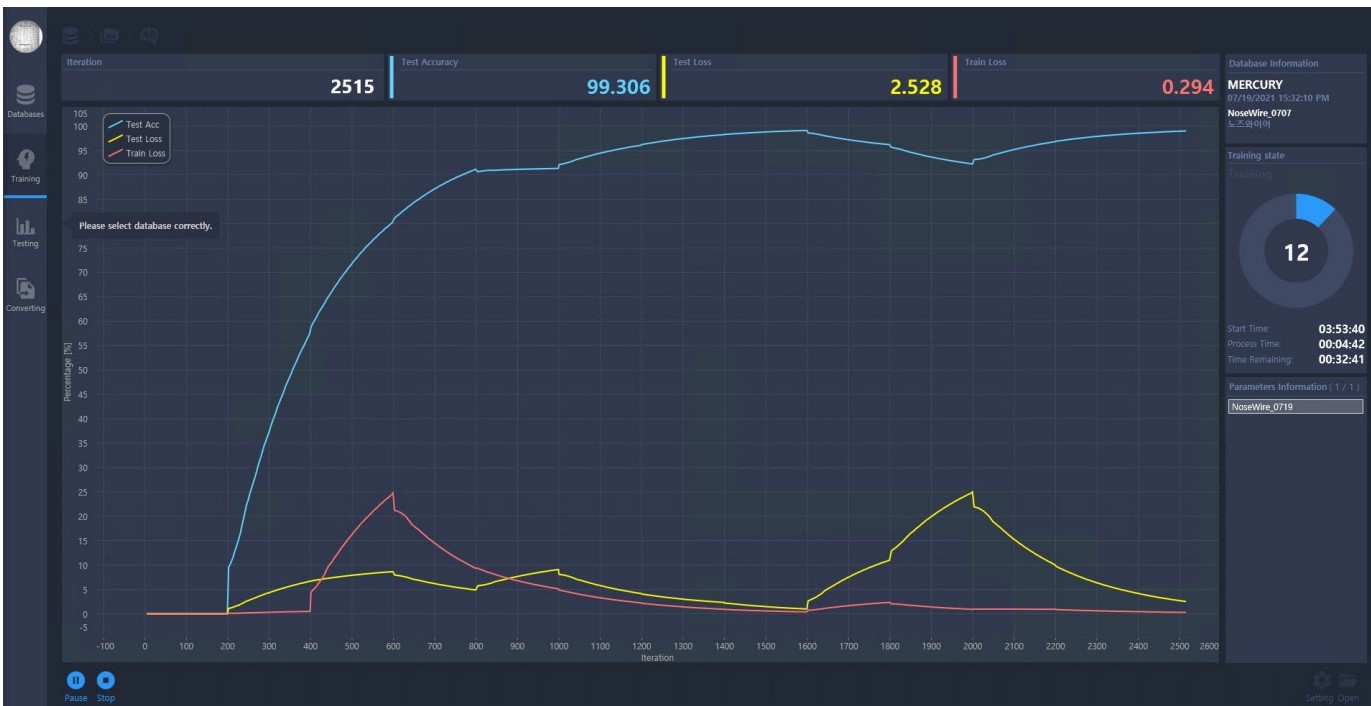

**Figure 17.** AI model training process on screen.

### 4.4. Results

Table 3 summarizes the results of performance evaluation using production, quality, cost, and delivery as KPIs to measure performance after the introduction of the Machine Vision-Based Quality Inspection System in the mask manufacturing process. The KPI and performance are verified by smart factory experts from the Ministry of SMEs and Startups and Korea Smart Manufacturing Office (KOSMO) following strict standards. In the evaluation of Table 3, 'before' refers to the situation before the establishment of the Machine Vision-based Quality Inspection System, and 'after' refers to the situation after the establishment of the system.

**Table 3.** Evaluation Indicator by KPI.

| # | Evaluation | KPI | Unit | Before | After | Note |
|---|---|---|---|---|---|---|
| 1 | P | Production per Hour | ea | 2600 | 4189 | 61.1% increase in productivity |
| 2 | Q | Reduce Defect rate | % | 8.3 | 5.69 | Decrease in average defective rate |
| 3 | C | Working time | Hour | 12 | 10.2 | Required time from input to ejection |
| 4 | D | Lead time from order to shipment | Hour | 38.5 | 23.87 | Based on shipment of 100,000 masks per equipment |

In the evaluation column, P represents Productivity. The hourly production of masks were previously 2600 sheets per hour, and after the establishment of the Machine Vision based Quality Inspection System, 4189 sheets per hour were produced, increasing the production by 61.1%. Q represents the Quality, which checks how much the defect rate has been reduced in the mask manufacturing process. In the existing mask manufacturing process, the defect rate averaged 8.3%, but after the Machine Vision-based Quality Inspection System was introduced, the defect rate decreased to 5.69%. C stands for Cost. This estimates how much cost occurs based on the number of hours workers work in the factory. In the existing mask manufacturing process, workers worked 12 h a day, but after construction, only 10 h of work satisfies daily production. D stands for Delivery. To confirm this, we check how long it takes based on the lead time from order to shipment. In the existing mask manufacturing process, it took 38.5 h to produce and ship 100,000 masks, but

after the establishment of the Machine Vision-based Quality Inspection System, the same delivery can be completed in 25 h.

Table 4 expresses the logic for calculating each KPI and formulates it. Production refers to the amount of production per hour and is calculated by dividing the production by the input time. Quality represents the defect rate occurring in a finished product, which is calculated by dividing the number of productions by the number of defects occurring in the finished product and multiplying it by 100. Cost is the time spent on the task to numerically represent the work capabilities that the worker can produce in the mask production task. Delivery calculates the time it takes to manufacture and ship products from receipt of orders. To this end, 100,000 masks were divided into daily production of equipment, focusing on the fact that they can be shipped in units of 100,000 masks.

**Table 4.** The Basis for Calculating KPI.

| KPI | Definition | Calculation Formula |
|---|---|---|
| Production | Production Per Hour(ea) | $\frac{Mask Manufactured}{Input hours}$ |
| Quality | Defect rate from finished product | $\frac{Defects}{Total products} \times 100\%$ |
| Cost | Time spent on work and Working time | n/a |
| Delivery | Lead time from order to shipment | $\frac{100,000 \; ea per equipment}{Daily Production per equipment}$ [1] |

[1] Mask manufacturers in the same industry produce an average of 100,000 masks a day.

Table 5 summarizes the Qualitative Performance after the introduction of the Machine Vision-Based Quality Inspection System evaluated from the perspective of Factory Workers and Management Workers. This qualitative performance measurement was conducted by surveying workers at the mask production site and office workers in charge of business management, and it thoroughly complied with the anonymity of the respondents' evaluations to increase reliability. For Factory Workers, the main concerns are quality improvement using Sensor and Machine Vision, Adaptation of Control/Inspection Automation, and Real-time Data Management. Target Achievement in the Quality Improvement category is to reduce worker fatigue through Sensor Control, increase inspection accuracy through the introduction of Machine Vision, and improve quality through data analysis. In the case of Adoption of Control/Inspection Automation, efficient use of time through the automation, real-time manufacturing site control through data-based feedback, and real-time operation monitoring can be performed in real time. Real-time Data Management is to improve real-time production management through computerization.

On the other hand, the main interests of Management Workers are Quality Improvement, Management Planning, and KPI (Key Performance Indicator). Quality Improvement is to enhance manufacturing capacity by using sensor data, improve quality assurance through the introduction of machine vision, and improve the production process to ultimately make a good mask and amplify the customer's experience. Management planning executes production planning through data collection in the manufacturing process, secures base data in smart factory construction, creates a visible management system by monitoring the production site, and secures a quality management system through data collection. KPI focuses on real-time quality control implementation, goal-oriented on-site data processing, and improved quality control using computer equipment.

Table 6 is a comparative chart that compares and analyzes the effects of the Machine Vision-Based Quality Inspection System (After) compared to the existing system (Before) in each process. Process includes the main inspection processes; Fabric Tension Control, Cutting of the Nose Wire Insertion, Ultrasonic Welding and Alignment Foreign Matter Inspection, Automation of final inspection process, Manufacturing facility management, Real-time manufacturing history management, and Production status monitoring. Before describes problems in the existing inspection process and includes actual photos at

the inspection process site. After also includes actual photos of each process after the introduction of the Machine Vision System and describes improvements. Effectiveness details what effectiveness and performance improvements have actually been achieved by comparing Before and After, and details the operations received from the field workers and management workers of Machine Vision System.

**Table 5.** Qualitative Performance.

| Point of View | Item | Target Achievement |
|---|---|---|
| Factory Workers | Quality Improvement (Sensor, Machine Vision) | Employee fatigue from constant monitoring is reduced because fabric input is controlled based on sensor control |
| | | Improved product inspection accuracy by adopting Machine Vision |
| | | Quality improvement through data analysis |
| | Adoption of Control/ Inspection Automation | Enabled efficient use of time through the automation |
| | | Enhanced manufacturing control ability through real time data feedback |
| | Real-time Data Management | Better initial responsiveness by real time operation monitoring |
| | | Improved real-time production management by the computerization |
| | | Enabled quick decision making by real-time data analysis |
| Mgmt. Workers | Quality Improvement (Sensor, Vision) | Strengthened manufacturing capacity based on sensor data feedback |
| | | Improved quality assurance by the introduction of machine vision system |
| | | Boosted customer experience by improving manufacturing process |
| | Management Planning | Implementation of manufacturing plan through manufacturing process data collection |
| | | Securing base data for smart factory construction |
| | | Transition to a visible management system by monitoring production sites |
| | | Securing quality management system through data collection |
| | | Real-time quality management by the computerized equipment |
| | KPI | Goal-oriented on-site data processing |
| | | Enhanced quality management through equipment improvement |

**Table 6.** Before vs. After Comparison.

| Process | Before | After | Effectiveness |
|---|---|---|---|
| Fabric Tension Control | Manual tension control unit | Since sensor data is used, the fabric can always maintain a constant processing speed through automatic control such as left and right alignment, weight, etc. | Prevention of defects caused by left and right alignment and tension |
| | Failure to automatically control tension depending on the weight of the fabric | The tension is controlled according to the weight of the fabric and the processing is maintained at a constant speed | At least 40% improvement in total defects |
| | The speed of fabric transfer varies due to the tension that changes over time after fabric replacement | Easy to check and correct the insertion and cutting time of the nose wire | Identified the cause of tension failure in the post-processing part |
| | The insertion position of the nose wire is different due to the failure of tension control | | |
| | Adjustment of left and right position of fabric is not automated | Prevent defects by maintaining the left and right alignment of the fabric | Losses due to poor processing of raw materials were reduced by resolving the causes of defects at an early stage |
| | Unnecessary fabric waste occurs because the fabric is checked at the finishing stage | | |

**Table 6.** *Cont.*

| Process | Before | After | Effectiveness |
| --- | --- | --- | --- |
| Cutting of the Nose Wire Insertion | A gap caused by a backlash through chain power transmission Shaking occurred from the chain and equipment when cutting Nose wire cutting time is not inserted at a certain point due to subtle differences | An independent cutting method in the form of an air cylinder Not affected by the shaking of the chain and equipment<br><br>The nose wire insertion time is kept constant | The nose wire insertion defect was improved in the part affected by the tension control unit |
| Ultrasonic Welding and Alignment Foreign Matter Inspection | No intermediate process inspection<br><br>Since left and right alignment defects are performed in the final inspection stage, there is fabric wasted due to raw material defects (20 times or more a day)<br><br>If the mask is opened without sampling inspection, it cannot be sold after a full inspection | Conducting internal inspection before entering the fabric folding process<br><br>Alignment and fusion shape inspection of left and right fabrics<br><br>The welding machine and the proximity sensor are interlocked in consideration of the transfer of fabric suitable for the rotational speed of the welding machine | Prevention of defects caused by left and right alignment defects and tension<br><br>At least 40% of total defects can be improved (up to 80%)<br><br>Fully inspected for internal foreign substances Defects caused by foreign substances can be detected |
| Automation of final inspection process | The experience of the inspector becomes the standard for judging good and defective products<br><br>Determination of good and defective products by visual inspection | Determination of good and defective products according to measurement results by machine vision inspection<br><br>Based on sensor data, facility setting values are adjusted to maintain quality | Managed inspection information for each product<br><br>Improved product quality analysis through real-time quality defect inspection and defect removal |
| Manufacturing facility management | No facility management records<br><br>No analysis on the cause of product defect | Real-time setting changes and maintenance to produce uniform quality products based on machine vision and sensor data<br><br>Gathering optimal mechanical setting data through collection and analysis of defective product judgment data | Secured manufacturing quality stability<br><br>The facility condition is checked according to changes in required values and automatic setting to maintain uniform quality by storing the facility setting history of the POP system |
| Real-time manufacturing history management | No history management<br><br>Real-time production is not confirmed Product shipment and production cannot be checked<br><br>The production volume of good and defective products is not counted | Real time manufacturing history can be managed through POP<br><br>Real-time utilization rate and production information can be checked | Securing product manufacturing real-time information<br><br>Capable of understanding production history and interworking with ERP<br><br>Product maintenance and quality control by manufacturing quality |
| Production status monitoring | Real-time production information is unknown | Real-time production in-formation monitoring | On-site management by dashboard Support quick decision making for management Quick response to problems |

## 5. Conclusions

To establish a sustainable production environment in the mask manufacturing process, machine vision-based quality inspection system, control automation, POP manufacturing monitoring system, smart factory hardware and software system, and data aggregation system by process and equipment were established. In addition, an image training model using LAON PEOPLE NAVI AI Toolkit is developed to classify good products and defective products with excellent performance in the mask manufacturing process. As a result, the productivity of "Company A"'s mask defect detection process can be dramatically increased.

An unresolved problem in this study is that in certain cases, inaccuracies in the detection of defective products may arise. In addition, it revealed that good products can indeed look like defective products at the time of detecting the image from the machine vision camera. Issues such as excess fabric scrap, fabric wrinkling, and fabric pulling are caused by the movement of the gripper. What this suggests is that even when machine vision systems are deployed in the field, human assistance is still necessary. This is because operator intervention is unavoidable for reasons such as fabric exhaustion and equipment maintenance in the manufacturing process. The challenge for future research is to find a machine vision construction methodology that resolves the unresolved inaccuracies mentioned above, and to apply the machine vision to other industries beyond the mask manufacturing process to dramatically increase PQCD and ensure a sustainable business environment.

**Author Contributions:** Conceptualization, M.P.; methodology, M.P.; validation, M.P. and J.J.; formal analysis M.P.; investigation, M.P.; resources, M.P.; data curation, M.P.; writing—original draft preparation, M.P.; writing—review and editing, M.P.; supervision, J.J.; project administration, M.P.; funding acquisition, J.J. All authors have read and agreed to the published version of the manuscript.

**Funding:** This research was supported by the MSIT (Ministry of Science and ICT), Korea, under the ICT Creative Consilience Program (IITP-2022-2020-0-01821) supervised by the IITP (Institute for Information & communications Technology Planning & Evaluation), and the National Research Foundation of Korea (NRF) grant funded by the Korea government (MSIT) (No. 2021R1F1A1060054).

**Institutional Review Board Statement:** Not applicable.

**Informed Consent Statement:** Not applicable.

**Data Availability Statement:** Not applicable.

**Acknowledgments:** This research was supported by the SungKyunKwan University and the BK21 456 FOUR (Graduate School Innovation) funded by the Ministry of Education (MOE, Korea) and National 457 Research Foundation of Korea (NRF).

**Conflicts of Interest:** The authors declare no conflict of interest.

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
