# Peer review of "Design and Implementation of Machine Vision-Based Quality Inspection System in Mask Manufacturing Process"

_sustainability, doi:10.3390/su14106009_

Round 1

Reviewer 1 Report

I recommend rejecting this paper because this paper reads like a technical report to introduce an implementation of a smart factory. It is difficult to find what is the own contributions from the author.

Author Response

Your kind and detailed comments have been a great help in revising the shortcomings of my thesis and revising the thesis in a better way. I express my sincere gratitude for this.

Many Thanks Indeed

Reviewer 2 Report

Minor remarks

  • All minor remarks are highlighted in the manuscript.
  • Author should choose the type of paper (Article, Review, Communication, etc.) and give the name of the institution.

Major remarks

The presentation of the manuscript is more like a report and it is not the usual way of a scientific paper should have. Also, no obvious originality can be identified from the current version of the presentation. Please investigate your work and clarify the originality. Then, please prepare your presentation into a scientific article that should solve some unsolved problems related to your study. Also, the results obtained in this study should be compared with the results available in the literature.

Author Response

(The authors gave the same response as above.)

Reviewer 3 Report

General Comments

  1. There are some typos or grammatical errors in the manuscript. Please check.
  2. In the manuscript, the authors use so many past-tense sentences which should be replaced with their present-tense mode.
  3. Please consider to replace [1, 4, 12] with more modern references or state-of-the-art reviews.

Suggestions

Line 10

The reference of [1] is improper here. Please move the reference of [1] to the Section 1.

Lines 56-54

Please give references to "LAON PEOPLE's NAVI AI program".

Line 239

quality management were ...

=> quality management is ...

Line 263

The authors wrote "... equipment. ip Time, ..."

=> ??

Author Response

(The authors gave the same response as above.)

Reviewer 4 Report

This paper presents a machine vision-based quality inspection system, with related equipment, in the mask manufacturing process.

The paper is clearly written but, unfortunately, is missing in some relevant aspects, as highlighted in the following.

SPECIFIC COMMENTS

  • In the state-of-the-art, the relevant issues of data augmentation, quality control, and inspection are addressed; however, the equally pertinent topic of "remaining useful life" was not addressed. The remaining useful life is relevant due to machines' huge impact on product quality. It is recommended to search literature by using the query string: remaining useful life 3d data, to update the state-of-the-art with studies regarding remaining useful life prediction from 3d data captured with advanced vision systems (i.e., profilometers, scanners, stereo/multi-vision, and so on).
  • The number of bibliographic references must be significantly increased.
  • What are the authors' contributions to the state-of-the-art? Comparing AS-IS and TO-BE is not enough for a scientific paper. The authors should clearly state their contributions to the advancement of the current state-of-the-art. They should also highlight the limitations of the current state-of-the-art addressed by their work.
  • The authors' contributions should produce measurable results that allow readers to appreciate the advancements compared with the current state-of-the-art.
  • Many acronyms are not defined in the manuscript, such as POP, OECD, MERS, COVID, etcetera. All acronyms should be defined the first time they are used.

LINE COMMENTS

Lines 56-57

 Quote: "LAON PEOPLE 's NAVI AI program"

 Comment: References to software manufacturer and datasheet should be provided.

Table 1

The authors should provide references about manufacturers and datasheets for all involved equipment.

Author Response

(The authors gave the same response as above.)

Reviewer 5 Report

Summary

This paper aims to present a machine vision-based quality inspection system that introduces additional equipment such as monitors and sensors to improve the productivity in the mask manufacturing process. While the paper provides detailed analysis in terms of hardware, the discussion in terms of specific computer vision models is unclear, and the claims are mostly based on qualitative description without clear experimental results, which prevent it from being accepted in the current form.

Major comments:
1. Section 2.2 and 2.3 of the related work are written vaguely. Not enough citations are included to support the claims and statements.

2. In Section 3.1, the motivation of the paper is unclear. Specifically, there are some terms that do not align with Figure 1 and need clear definitions or explanation, such as  "Fabric derivation", "contamination test", and "internal inspection test". It would be helpful to make better use of Figure 1 to show the specific steps in the workflow this work aims to solve.

3. In the methodology section, it is unclear which specific models this work uses. The specific library used in the manuscript, "LAON PEOPLE NAVI AI library" is not sufficiently discussed and is lack of introduction.

4. The abstract claims that "deep learning and machine vision-based anomaly detection experiments were performed", but the manuscript does not seem to provide the experimental data, or the specific configuration of the anomaly detection experiments. Table 4 shows the metrics used in the work, but limited numeric results are given, which makes it hard to justify the performance of the proposed model. In fact, the experimental section seems missing. It is insufficient to make claims by providing qualitative analysis only without experimental support.

5. The discussion of Table 5 and Table 6 is missing. Further explanation and analysis are recommended.

Author Response

(The authors gave the same response as above.)

Round 2

Reviewer 1 Report

I agree with the author's correction and recommend accepting the current version.

Author Response

Thank you very much for your interest in my humble research and for your loving advice. As a scholar, I will always be a helpful member of society by focusing on research humbly with a sense of calling.   Many Thanks Indeed! Minwoo Park

Reviewer 2 Report

The manuscript can be accepted in this form.

Author Response

(The authors gave the same response as above.)

Reviewer 4 Report

This reviewer is very sorry that the author did not adequately address points 3 and 4.
Point 3: The mere professional application of a methodology does not represent an advancement of the state-of-the-art.
Point 4: The author has not provided measurable and comparable results with the state-of-the-art.

Author Response

Point 3: The mere professional application of a methodology does not represent an advancement of the state-of-the-art. &

Point 4: The author has not provided measurable and comparable results with the state-of-the-art.

I agree with you to a great extent. My paper did not specifically suggest improvements compared to SOTA. This is because in the case of existing SOTA papers, there is no case of introducing them into the mask manufacturing process and presenting the results. The Novelty of this paper is that it realistically presents a construction methodology for practical SW and HW to build a sustainable mask manufacturing process that no one has tried before.

Reviewer 5 Report

The reviewer would like to thank the author's great efforts in improving the paper by addressing most of the comments from the previous submission. However, one of the major comments, comment 4 (or point 6 in the cover letter) is not addressed. Without comparable experiments and sufficient numerical results, it is hard to tell the performance improvement of the proposed method, and its academic contributions are hard to justify. Unless such experiments are included with an in-depth model description and analysis (model methodology), the reviewer would retain the same recommendation from the last round.

The reviewer would agree that the manuscript can be viewed as a technical report with a detailed description of the system hardware infrastructure and qualitative evaluation, but it cannot be viewed scientific publication in its current form.

Author Response

Point 1: Without comparable experiments and sufficient numerical results, it is hard to tell the performance improvement of the proposed method, and its academic contributions are hard to justify.

  • I totally agree with you. In the case of this paper, rather than analyzing quantitative results in time series order, we focus on actual performance improvement by comparing before and after smart factory construction.

Point 2:The reviewer would agree that the manuscript can be viewed as a technical report with a detailed description of the system hardware infrastructure and qualitative evaluation, but it cannot be viewed scientific publication in its current form.

  • I agree with you a lot. The form of this paper may be different from the typical Scientific paper form. However, this paper demonstrates visible performance in real-world factories with a thoroughly scientific and logical approach in SW as well as HW installation and deployment process to build Smart Factory in mask manufacturing process.
  • The methodology presented in this paper was strictly reviewed and finally approved by a professional member of the Ministry of SMEs and Startups in Korea Smart Manufacturing Office (KOSMO). For this reason, I think it is sufficient to be recognized for the justification of the performance improvement.

Round 3

Reviewer 4 Report

The authors did not address points 3 and 4 reported in the previous review round.

This reviewer is very sorry but does not agree with the publication of the manuscript in a scientific journal due to the severe shortcomings reported.

The reviewer recommends considering submitting in a journal with a professional slant focused on manufacturing processes.

Reviewer 5 Report

The reviewer would like to thank the authors' efforts in improving the paper. Given the fact that the manuscript has switched its position from a typical academic research paper, I would recommended the acceptance of the paper if it could explicitly state that it is conducted as an application study for a real-world factory for performance improvement with qualitative analysis at the beginning of the paper (the abstract and/or the introduction).
